# Probabilistic River Water Mapping from Landsat-8 Using the Support Vector Machine Method

**Qihang Liu [1], Chang Huang [1,2,3,\*], Zhuolin Shi [1] and Shiqiang Zhang [1,2]**

[1]  College of Urban and Environmental Sciences, Northwest University, Xi'an, Shaanxi 710127, China; lqh@stumail.nwu.edu.cn (Q.L.); shizl@stumail.nwu.edu.cn (Z.S.); zhangsq@nwu.edu.cn (S.Z.)
[2]  Shaanxi Key Laboratory of Earth Surface System and Environmental Carrying Capacity, Northwest University, Xi'an, Shaanxi 710127, China
[3]  Institute of Earth Surface System and Hazards, Northwest University, Xi'an, Shaanxi 710127, China
\*  Correspondence: changh@nwu.edu.cn; Tel.: +86-29-8830-8412

**Abstract:** River water extent is essential for river hydrological surveys. Traditional methods for river water mapping often result in significant uncertainties. This paper proposes a support vector machine (SVM)-based river water mapping method that can quantify the extraction uncertainties simultaneously. Five specific bands of Landsat-8 Operational Land Imager (OLI) data were selected to construct the feature set. Considering the effect of terrain, a widely used terrain index called height above nearest drainage, calculated from the 1 arc-second Shuttle Radar Topography Mission (SRTM) digital elevation model (DEM), was also added into the feature set. With this feature set, a posterior probability SVM model was established to extract river water bodies and quantify the uncertainty with posterior probabilities. Three river sections in Northwestern China were selected as the case study areas, considering their different river characteristics and geographical environment. Then, the reliability and stability of the proposed method were evaluated through comparisons with the traditional Normalized Difference Water Index (NDWI) and modified NDWI (mNDWI) methods and validated with higher-resolution Sentinel-2 images. It was found that resultant probability maps obtained by the proposed SVM method achieved generally high accuracy with a weighted root mean square difference of less than 0.1. Other accuracy indices including the Kappa coefficient and critical success index also suggest that the proposed method outperformed the traditional water index methods in terms of river mapping accuracy and thresholding stability. Finally, the proposed method resulted in the ability to separate water bodies from hill shades more easily, ensuring more reliable river water mapping in mountainous regions.

**Keywords:** landsat operational land imager (OLI); support vector machine (SVM); height above nearest drainage (HAND); normalized difference water index (NDWI); posterior probability

## 1. Introduction

Water area variation is the most direct reflection of river water regime dynamics. River width, which could be easily extracted from the river water area, is a key input of river hydrodynamic models and a core parameter for estimating river discharge [1]. Therefore, there is an increasing demand for accurate river water area measurements from remote-sensing images, including optical images and synthetic-aperture radar (SAR) images. Numerous studies have proposed a series of methods to map river water, including several water index methods [2–5] and thresholding methods [6,7]. However, all of these methods inevitably cause uncertainties [8]. For example, uncertainty often occurs when identifying water pixels by water index values on optical images as it is often difficult to determine whether a pixel denotes water or non-water when its water index value is near the

segmentation threshold [9]. The reason for this is twofold: the mixed pixels and the quality of the remote-sensing images. To account for these uncertainties, studies on mixed pixel decomposition and reconstruction regarding water mapping at a subpixel level have been conducted [10] and found to be effective [11] to a certain extent. Nevertheless, the spectral decomposition models and subpixel mapping methods still create a series of uncertainties. Some properties of remote-sensing images (SAR images in particular), such as imaging mode and resolution, would make the classification results uncertain [12]. Moreover, in mountainous areas, the effect of the terrain is problematic if it either blocks the emission of microwave signals or confuses water detection with hill shades on optical images. For optical images, cloud coverage is another issue that often affects extracting the water area.

Therefore, more attention has been paid to the uncertainty in extracting surface water from remote-sensing data. For example, Refice et al. proposed a semi-automatic method to estimate the posterior probability of flooding using SAR images and auxiliary data [13]. Westerhoff et al. constructed a probability model based on the backscatter values and angles of incidence of a long-term training sample dataset to estimate the conditional probability for each SAR pixel that consisted of the water body [14]. Giustarini et al. obtained the probability distribution function of backscattering values associated with water bodies and non-water bodies based on the backscatter histogram decomposition of SAR images, then used this function to estimate the probability that each pixel was part of the water body [15]. All the aforementioned studies are based on SAR images. For optical images, Jia et al. explored the probability of Landsat pixels as water bodies based on a spectral matching method and then employed a particle swarm optimization method to achieve the best interpretation of water bodies [16]. However, the spectral matching method does not account for shadows and clouds well, which limits its application in mountainous areas.

Probability is a common means of quantifying uncertainties. However, due to the different principles of the above methods, the measurement scale and the accuracy of their results are different from each other. Therefore, it is difficult to ensure the consistency of the derived probability of water area, and limits further improvement on river water mapping through fusing multisource remote-sensing data to reduce uncertainties. Therefore, this study aims to propose a posterior probability support vector machine (PPSVM) method, which is a unified framework for river water mapping in mountainous areas and, in the meantime, quantify the uncertainties with probabilities.

The remainder of this paper is organized as follows: Section 2 includes the introduction of the study area and data; Section 3 describes the methodology; Section 4 provides a discussion regarding the optimization, stability and accuracy of the proposed method; and Section 5 presents some concluding remarks.

## 2. Study Area and Data

### 2.1. Study Area

Three river sections with different geographical settings and diverse water characteristics were selected as the case study areas; these comprised Zhengyixia and Yingluoxia of the Heihe River and Shuangtabu of the Shule River (Figure 1). Zhengyixia is located in the middle and lower reach of the Heihe River and has a large number of sandbanks and bifurcations, which makes river water extraction difficult. Shuangtabu is located downstream of the Shule River and has meandering, narrow, and flattened river channels, accompanied by rich floodplain wetlands. Yingluoxia is located in the middle and upper stream of the Heihe River, which is a mountainous area with broad hill shade. The selected river sections include the typical situations that make it difficult to conduct river water mapping from remote-sensing data. To better demonstrate the water mapping details at some key areas, nine demonstration zones, as shown in red rectangles in Figure 1, were selected. The width of these river sections varies from about 30 m to 100 m. River water mapping was conducted across the extent of each study area, and the results will be displayed with special regard to these demonstration zones.

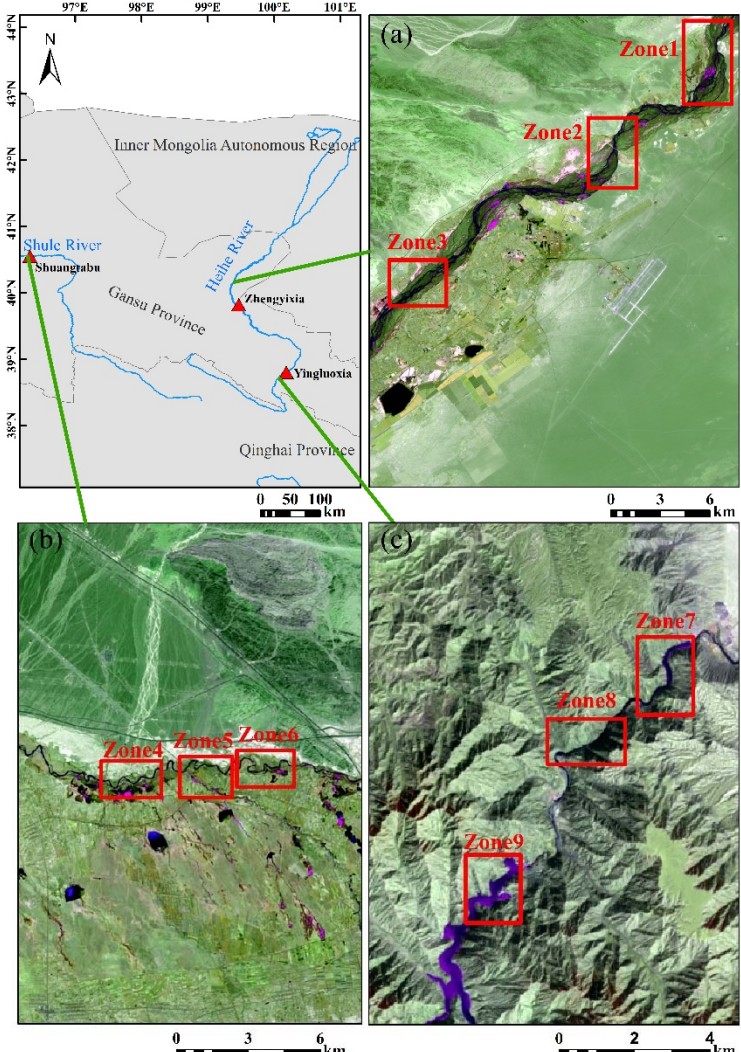

**Figure 1.** Three study areas (**a**) Zhengyixia reach of the Heihe River, (**b**) Shuangtabu reach of the Shule River, (**c**) Yingluoxia reach of the Heihe River, and nine demonstration zones (in red rectangles).

## 2.2. Data

In this study, river water mapping was conducted on Landsat-8 Operational Land Imager (OLI) images. For each site, an image of Landsat-8 surface reflectance product was collected from Google Earth Engine (GEE) and used as the input to the support vector machine (SVM) classifier (Table 1). A 1 arc-second resolution Shuttle Radar Topography Mission (SRTM) and a digital elevation model (DEM) were collected from GEE and employed as the auxiliary data to generate the height above the nearest drainage (HAND) index [17,18]. The Sentinel-2A/2B Top Of Atmosphere (TOA) reflectance products from GEE acquired close to Landsat-8 images (Table 1) were employed as the reference data source for validating the river mapping results, considering their relatively higher spatial resolution (10 m for several bands, including the near-infrared band). A strict visual interpretation was conducted carefully on Sentinel-2 images, assisted with Google Earth images, to generate reliable 10 m resolution water maps as the reference. This visual interpretation process was carried out independently to our water mapping results, or the Landsat images in use.

**Table 1.** Landsat-8 and Sentinel-2 images used in this study.

| Site | Path/Row of Landsat-8 Image | Date of Landsat-8 Image | Date of Sentinel-2 Image |
|------|------|------|------|
| Zhengyixia | 134/32 | 2019-3-23 | 2019-3-22 |
| Shuangtabu | 136/32 | 2019-3-21 | 2019-3-18 |
| Yingluoxia | 134/33 | 2019-3-23 | 2019-3-22 |

## 3. Methods

Considering that there is usually cloud coverage on optical images like Landsat OLI, we first used a popular FMask algorithm [19] to identify those cloud and cloud shadow pixels and marked them as untrusted pixels to avoid confusion of the water pixel extraction. At the same time, the FMask could also output a set of pure land and pure water pixels, which were then employed as the training samples. A feature set was established by selecting bands from the OLI image based on the surface reflection ranges of pure water and land samples. A HAND image derived from SRTM DEM was also adopted into the feature set. Using this feature set, a posterior probability SVM model was constructed to identify water pixels together with posterior probability to quantify the uncertainties. Finally, a histogram-based threshold was conducted to convert the probability maps into binary water maps. Accuracy was evaluated by comparing these water maps with visually interpreted Sentinel-2 images. The merit of the proposed method was highlighted by comparing the traditional Normalized Difference Water Index (NDWI) and modified NDWI (mNDWI) methods regarding accuracy and stability. The flowchart of the methodology is shown in Figure 2.

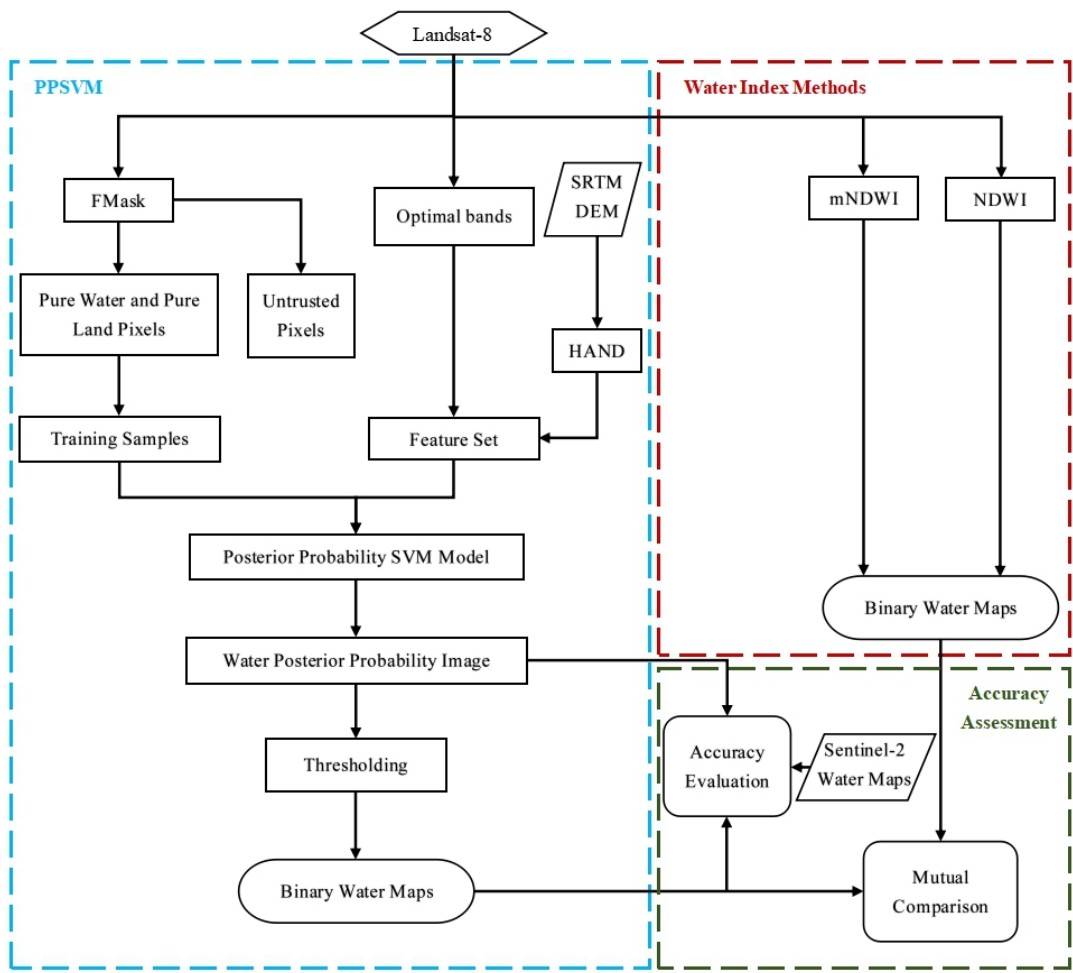

**Figure 2.** Flowchart of the methodology.

### 3.1. FMask Algorithm

The FMask algorithm was originally developed to identify the cloud, cloud shadow, and snow pixels on Landsat 4-7 images [19]. Using an object-based cloud and cloud shadow-matching algorithm make it possible to provide cloud, cloud shadow, and snow masks for each image, and has been widely used in the processing of Landsat images. Zhu et al. improved the accuracy of the algorithm to make it more suitable for Landsat-8 images. The modified algorithm also maintains high computation efficiency like the previous version [20].

This paper uses the modified FMask algorithm to identify cloud and cloud shadow pixels and mark them as untrusted pixels. The FMask method mainly extracts cloud and cloud shadow pixels by calculating their probability of presence. The new FMask algorithm [20] increases the calculation of cirrus cloud probability based on the original algorithm, which provides better cloud probability, especially for places with a large number of thin cirrus clouds.

These untrusted pixels will not be identified later by the river mapping methods due to information loss under cloud coverage. In the meantime, a set of ensured pure land and water pixels without interference have been obtained by the FMask algorithm. They will later be used as the training samples.

### 3.2. Feature Set Construction

As shown in Table 2, there are nine bands in a Landsat OLI image. To improve computation efficiency concentrating on the task of river water extraction, we made a selection on the features (bands) to be added into the feature set for the SVM model first. A boxplot chart (Figure 3) was made for seven bands of OLI, using the pure land and water samples derived from the FMask. It was clear that pure land and water samples exhibited a clear reflectance difference on the green band, red band, near-infrared (NIR) band, and two shortwave infrared (SWIR) bands. These bands were usually used to construct water indices, such as the NDWI and mNDWI. Therefore, these five bands were selected as features and added into the feature set.

**Table 2.** Investigated bands of Landsat Operational Land Imager (OLI) image.

| Name | Wavelength (μm) | Description |
|---|---|---|
| U-BLUE | 0.435–0.451 | Band 1 (ultra blue) surface reflectance |
| BLUE | 0.452–0.512 | Band 2 (blue) surface reflectance |
| GREEN | 0.533–0.590 | Band 3 (green) surface reflectance |
| RED | 0.636–0.673 | Band 4 (red) surface reflectance |
| NIR | 0.851–0.879 | Band 5 (near infrared) surface reflectance |
| SWIR1 | 1.566–1.651 | Band 6 (shortwave infrared 1) surface reflectance |
| SWIR2 | 2.107–2.294 | Band 7 (shortwave infrared 2) surface reflectance |

Considering the close relationship between river water distribution and terrain, we assumed that the terrain data was helpful for identifying the river water. Therefore, a HAND index image derived from SRTM DEM data was supplemented into the feature set. The HAND index was generated using two sets of procedures. The first was used to condition the input DEM data by fixing sinks, defining flow paths, calculating an accumulated area map, and defining the drainage networks based on the accumulation map. The second procedure was to use the local drain directions and the drainage network to generate the nearest drainage map. Each pixel on this map is spatially associated with all the DEM pixels draining into it. Then, each DEM pixel should have an elevation difference with its associated nearest drainage pixel. This elevation difference was assigned as the HAND index value for the DEM pixel. A detailed description of generating a HAND image from DEM can be referred to in [21]. The HAND index maintains the height value above the nearest drainage. Compared with the original elevation, the HAND value was more closely related to the probability of the water presence.

Therefore, six features, including five bands of OLI (green, red, NIR, SWIR1, SWIR2), which were sensitive to the water and HAND index image derived from 1 arc-second SRTM DEM, were selected to construct a feature set for the SVM classifier.

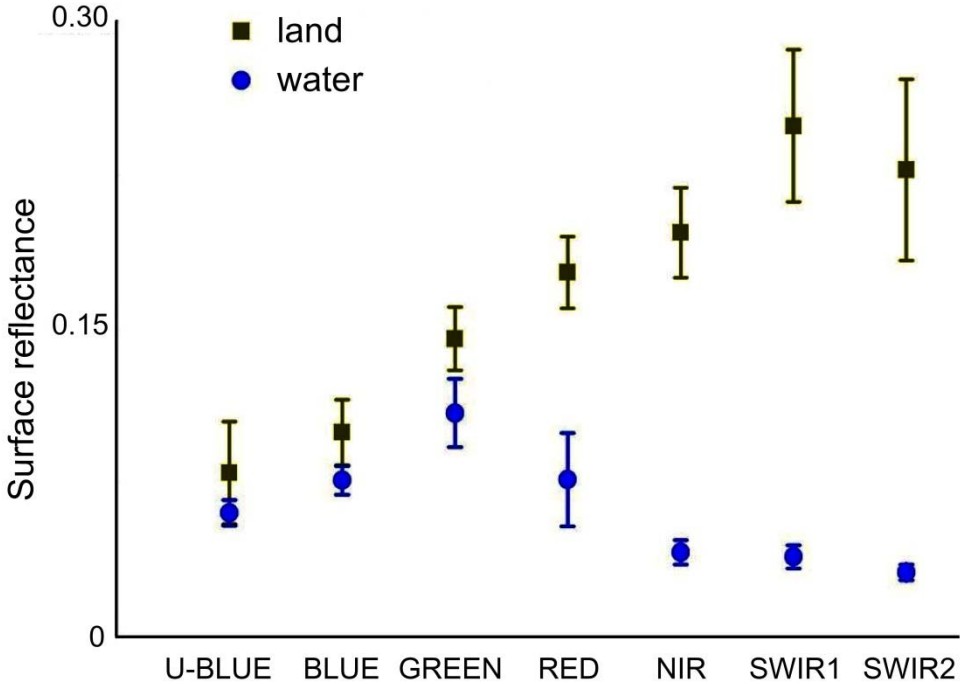

**Figure 3.** Distribution of the surface reflectance ranges of pure water and land samples on Landsat-8 OLI bands.

### 3.3. Posterior Probability Support Vector Machine (SVM)

SVM, one of the supervised learning models [22], has been implemented in many fields, including pattern recognition, classification, and regression analysis. The feature set in SVM is a group of vectors, which may be numeric, categorical, or logical. Each element in the group specifies a corresponding row of the training samples [23]. The main advantage of SVM is its classification ability to solve a non-linear problem using different kernel functions. It has been widely used for binary classification. Originally, the idea of the SVM was to find a hyperplane to separate the two classes with the maximum margin. Water mapping was a typical binary classification problem that classified remote-sensing image pixels into water and non-water.

In this study, the six features in the feature set were employed to construct an SVM model for river water mapping. Set $T_{xy} = [(Green_1, Red_1, NIR_1, SWIR1_1, SWIR2_1, HAND_1), (Green_2, Red_2, NIR_2, SWIR1_2, SWIR2_2, HAND_2), \ldots, (Green_n, Red_n, NIR_n, SWIR1_n, SWIR2_n, HAND_n)]$ as the training sample set with $n$ samples. $x_i$ represents the $i$th vector of the training set. $yi \in [1, 2, 3, 4, 5, 6]$ represents the six features in each vector.

The SVM's discriminant function is:

$$f(x) = \sum_{i}^{N} y_i a_i K(x, x_i) + b \tag{1}$$

where $a_i$ is the Lagrange multiplier, $K(x,x_i)$ is the kernel function, and $b$ is the classification threshold. An SVM with different kernel functions can generate different algorithms [24]. The general verdict output of SVM is:

$$D(x) = \begin{cases} 1, f(x) > 0 \\ 2, f(x) < 0 \end{cases} \tag{2}$$

To illustrate a result of water or non-water, D(x) gives the values 1 or 2 after the classification, representing water or non-water, respectively. Traditional SVM models will not give any estimation about the uncertainties of their classification. However, a Platt transformation can be applied to derive the posterior probability of each pixel being identified as water. The Platt transformation for calculating posterior probability [25] is as in Equation (3).

$$P_{Y|x}(Y = water|f, \theta) = \frac{1}{1 + \exp(Af(x) + B)} \tag{3}$$

where $P_{Y|x}(Y = water|f, \theta)$ represents the probability value of the given pixel x belonging to the water class. The parameter vector $\theta = [A, B]^T$ may be obtained by the maximum likelihood estimation based on the training set as in Equation (4).

$$\theta = argmax_{\theta'} L(\theta'|T_{xy}) \tag{4}$$

where L is a log-likelihood function as in Equation (5).

$$L(\theta|T_{xy}) = \sum_{i=1}^{k} logp_{Y|x}(y_i|f(x_i), \theta) \tag{5}$$

Subsequently, the obtained parameters A and B are substituted into the Equation (3) to arrive at the posterior probability value of the pixels belonging to the water classification.

### 3.4. Water Mapping Using Traditional Water Index Methods

There are many methods that have been developed for identifying water pixels from optical remote-sensing imagery. For a comprehensive review, please refer to [8]. Water index is an easy but effective approach for water mapping. Many water indices, such as NDWI [2] (Equation (6)), mNDWI [3] (Equation (7)), have been widely used for this purpose. It was found that different water indices have different characteristics, none of which are superior to another [8]. This study selected two popular water indices (NDWI and mNDWI) as representatives for the traditional water mapping methods. After the water index images were derived using the corresponding bands, a threshold must be determined to segment the water index image into a binary water map.

$$NDWI = \frac{Green - NIR}{Green + NIR} \tag{6}$$

$$mNDWI = \frac{Green - MIR}{Green + MIR} \tag{7}$$

In Equations (6) and (7), Green refers to green band, NIR refers to near infrared band, and MIR refers to mid-infrared band (SWIR1 band in Table 2 was used here).

### 3.5. Accuracy Assessment

3.5.1. Accuracy Assessment on Posterior Probability Images

The accuracy of the posterior probability was evaluated by calculating the weighted root mean square difference ($E_{wrms}$) between the posterior probability image and the reference classification map. Specific steps are as follows [15]:

First, the probability values [0, 1] on the posterior probability image were divided into a certain number (N) of small and equal intervals. In this study, we set N = 10, and the intervals U were [0, 0.1], [0.1, 0.2],..., [0.9, 1.0].

Second, for each probability interval $U_i$, the number of pixels ($A_i$) that were observed as river water in the binary map from Sentinel-2 images was calculated, and $T_i$ is the total number of pixels in $U_i$. Then, the fraction ($F_i$) of the actual river water pixels in $U_i$ could be calculated by dividing $A_i$ by $T_i$.

Third, with an average probability value ($V_i$) in each interval, $E_{wrms}$ was calculated using Equation (8):

$$E_{wrms} = \sqrt{\frac{\sum_{i=1}^{N} (V_i - F_i)^2 T_i}{\sum_{i=1}^{N} T_i}} \tag{8}$$

The $E_{wrms}$ represents the proximity of the posterior probability distribution to the reference water pixel distribution. An $E_{wrms}$ value closer to 0 represented higher accuracy in the posterior probability image.

### 3.5.2. Accuracy Assessment on Binary Water Maps

Binary water maps were derived by thresholding the probability images using threshold values determined by the histograms. A series of popular accuracy indicators, including overall accuracy, commission error, omission error, the Kappa coefficient [26], and critical success index (CSI) were employed to quantify the accuracy of these binary water maps obtained by probability images through thresholding; 10 m resolution water maps were visually interpreted from Sentinel-2 images and used for reference. The resulting binary maps were first resampled to a 10 m resolution using the nearest resampling method before being compared with the reference.

Using A as the number of pixels that are water both in the resultant map and reference map, B as the number of pixels that are water in the reference map but land in the resultant map, C as the number of pixels that are land in the reference map but water in the resultant map, and D as the number of pixels that are land both in the resultant map and reference map, overall accuracy was calculated as the proportion of correctly classified pixels to the total number of pixels (Equation (9)):

$$\text{Overall accuracy} = (A + D)/(A + B + C + D) \tag{9}$$

Commission error was calculated as the percentage of the pixels that were misclassified as water (Equation (10)):

$$\text{Commission error} = C/(A + B + C + D) \tag{10}$$

Omission error was calculated as the percentage of the pixels that was misclassified as non-water (Equation (11)):

$$\text{Omission error} = B/(A + B + C + D) \tag{11}$$

The Kappa coefficient is a measure of classification accuracy calculated as Equation (12), where $P_0$ is overall accuracy, and $P_1$ was calculated as Equation (13). Here, N is the total number of the pixels, $T_{water}$ and $T_{land}$ is the number of water and land pixels in the reference map, $R_{water}$ and $R_{land}$ is the number of water and land pixels in the resultant map.

$$\text{Kappa} = \frac{P_0 - P_1}{1 - P_1} \tag{12}$$

$$P_1 = \frac{T_{water} \times R_{water} + T_{land} \times R_{land}}{N^2} \tag{13}$$

The critical success index (CSI) is an indicator for evaluating the accuracy of classification, which is less affected by an exaggeration of the non-water pixels (Equation (14)):

$$\text{CSI} = A/(A + B + C) \tag{14}$$

## 4. Result and Discussion

### 4.1. Posterior Probability Results

Figure 4a shows the river water mapping results in forms of posterior probability maps for the Zhengyixia reach. A referencing river water map for this reach was derived by visually interpreting the same day Sentinel-2 images was displayed as in Figure 4b. To reveal more details, three river sections (zones 1–3) were enlarged for display. From these enlarged sections, it was clear that pixels with higher probability values were concentrated in the center of the river channels, and the pixels with lower probability values were distributed around the river channels. Overall, Figure 4a,b have a high degree of matching, even in some small tributaries. Small differences still exist, such as some tributaries in Zone 3. In this area, there are some land pixels in the Sentinel-2 image but with high water probability values. This is likely due to the relatively low spatial resolution of the DEM data used in this study.

Then, based on Figure 4a,b, we calculated the weighted root mean square difference ($E_{wrms}$) between the posterior probability image and the referencing classification map. The value of $E_{wrms}$ was 0.067, indicating that the posterior probability results in the Zhengyixia reach were reasonable.

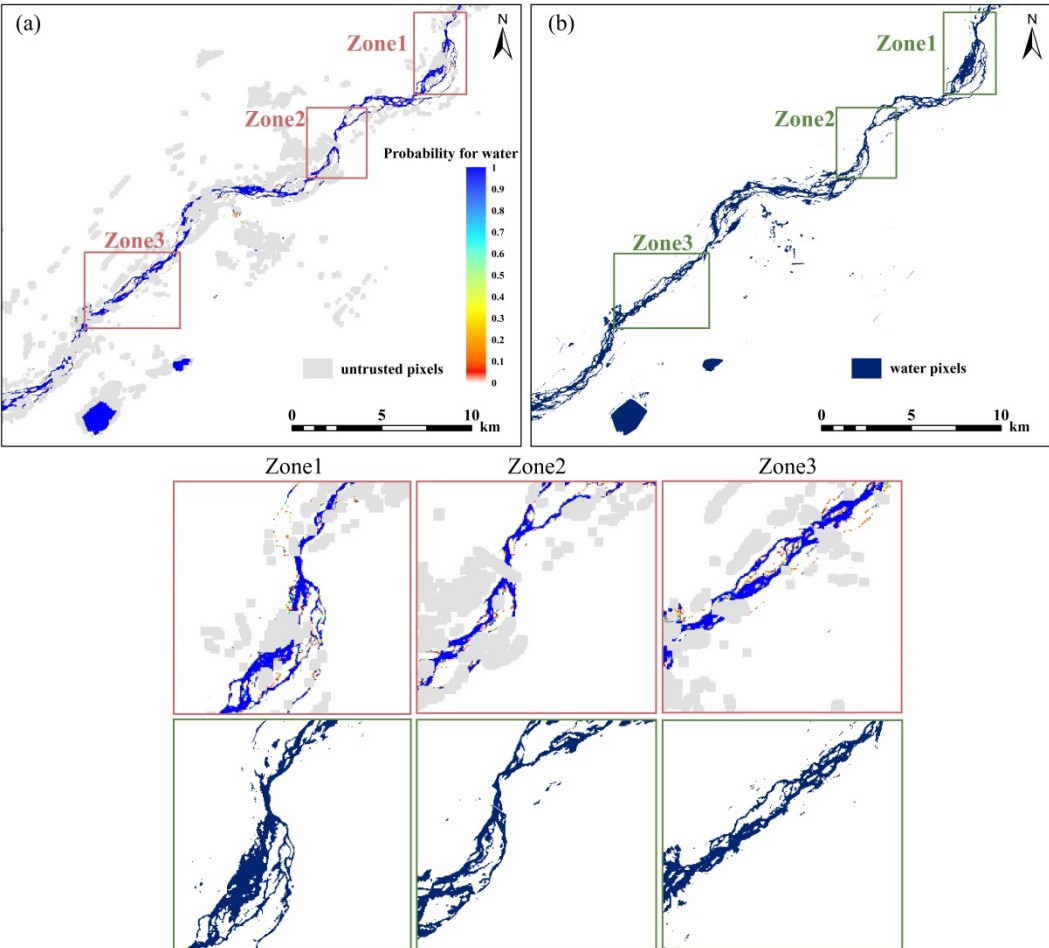

**Figure 4.** (**a**) The posterior probability map in the Zhengyixia reach of the Heihe River, and (**b**) the referencing water map derived from visually interpreting the Sentinel-2 image.

For the Shuangtabu reach, the resulting posterior probability map (Figure 5a) and the Sentinel-2 derived referencing map (Figure 5b) exhibit an even higher similarity since the river channels here are simpler than the Zhengyixia reach. However, this study area contains meandering river channels and wetlands. It is clear from the enlarged maps that the pixels with the high probability values are mainly distributed in the center of the river channels, but in some places, such as some turning corners of

the river, the probability of the water pixels may be lower. Overall, the resulting probability image in Zhengyixia reach is highly similar to the referencing water map. The $E_{wrms}$ of this study area is 0.057, smaller than that of the Zhengyixia reach, indicating an even higher accuracy.

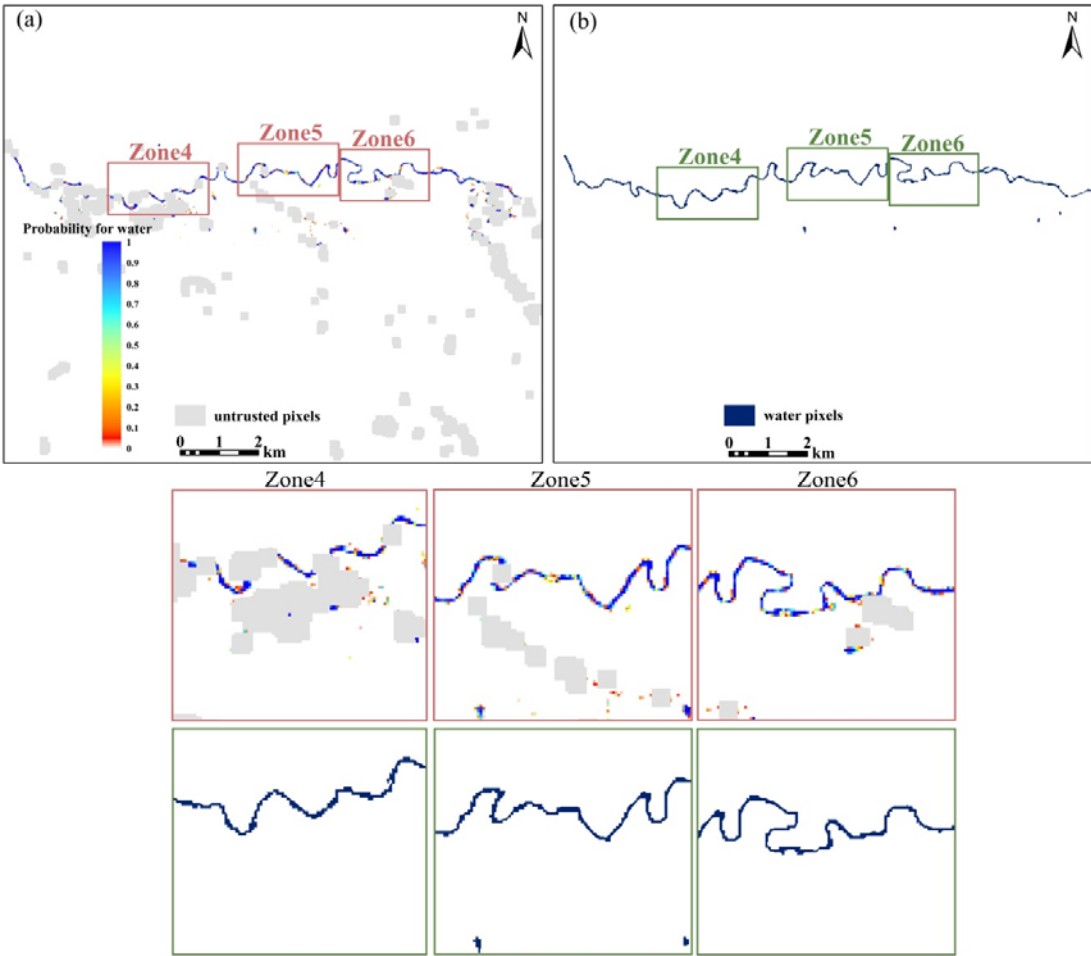

**Figure 5.** (**a**) The posterior probability map in the Shuangtabu reach of the Shule River, and (**b**) the referencing water map derived by visually interpreting the Sentinel-2 image.

The resulting probability map and referencing water map of the Yingluoxia reach are shown in Figure 6. Except for the water pixels with high probabilities of being water, there were some areas that were non-water as the Sentinel-2 had probability values greater than 0, even though these probability values were generally smaller than 0.5. Through a review of the original image, it was found that these areas are mostly hill shade areas. Hill shade pixels on Landsat-8 have similar reflectance as water pixels, which usually confuses water extraction. Involving the HAND index, the proposed method can distinguish these two types of pixels, with hill shade pixels having obviously low probabilities (approximately 0.2–0.3). The $E_{wrms}$ value for the Yingluoxia reach is 0.031, representing accuracy.

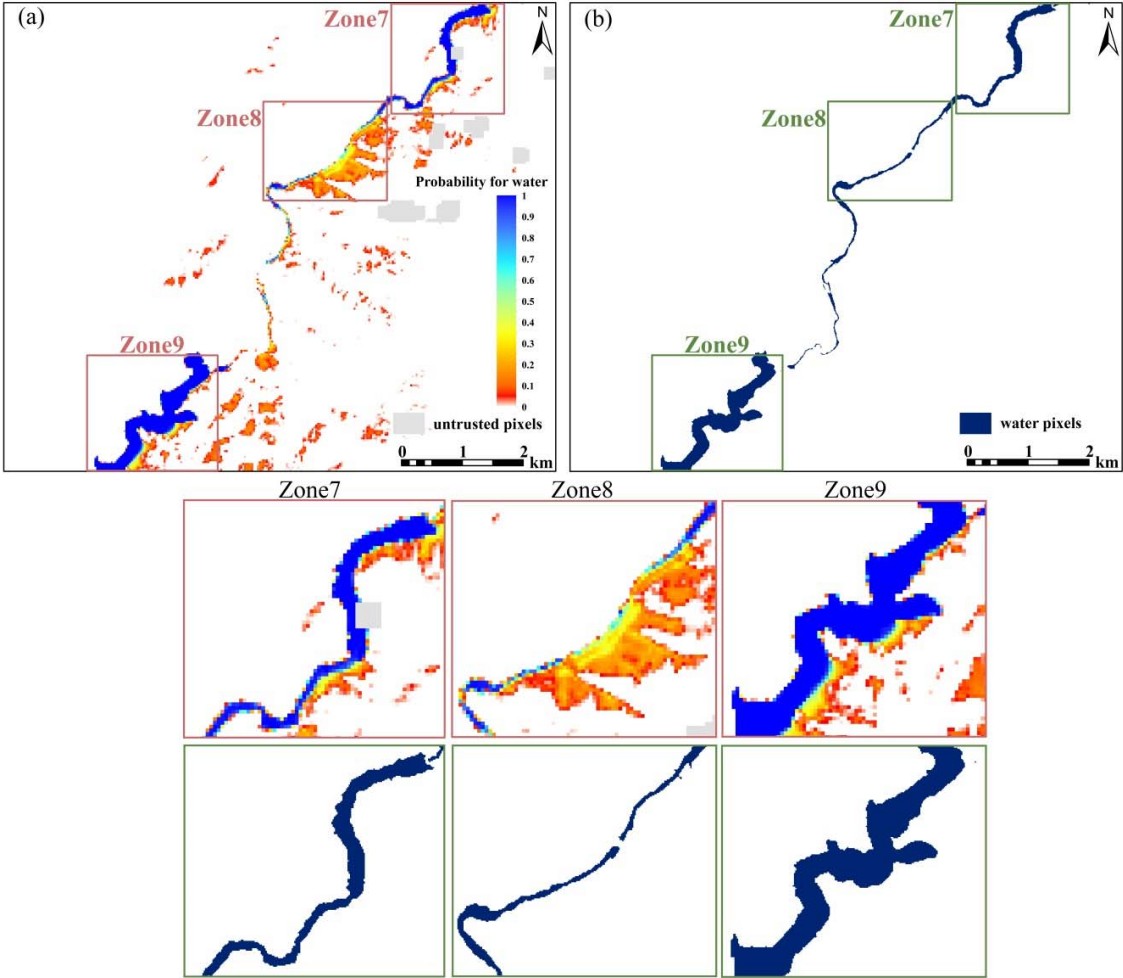

**Figure 6.** (**a**) The posterior probability map in the Yingluoxia reach of the Heihe River and (**b**) the referencing water map derived by visually interpreting Sentinel-2 images.

Figure 7 shows the boxplots of the posterior probability values in the referencing water areas of the three study areas. The median values of the probability (red line) were all near 1. For the referencing water areas, their probability values were distributed in a concentrated way, especially for the Zhengyixia reach, although there were many low outliers. A large number of outliers was due to the large extent of this study area. It can be observed from Figure 7 that the upper quartile and maximum indicators are both nearly overlapped by the median indicator. This suggests that in all study areas, our proposed method identified those true-water pixels as water with a high probability. Over half of these pixels had probabilities greater than 0.999, 0.998, and 0.999 in these three study areas.

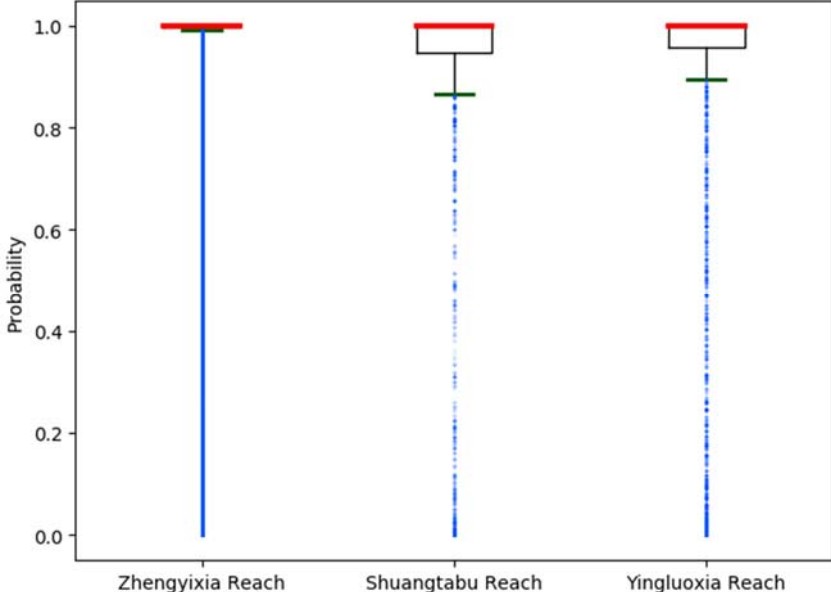

**Figure 7.** The boxplots of probability values obtained by posterior probability support vector machine (PPSVM) in the referencing water areas.

### 4.2. Histogram-Based Thresholding

In addition, NDWI and mNDWI are two widely used water-mapping methods. They are both simple and effective. However, an issue occurs in that an optimal threshold must be carefully determined [8] to segment the index images to the binary water maps as the threshold directly affects the accuracy of the water maps. In this study, the resulting posterior probability maps also required a threshold to be transferred into binary water maps. This process was compared with the traditional water index methods regarding the stability of the thresholding as well as the accuracy of the segmentation.

Figure 8 shows the histograms of the posterior probability images and mNDWI and NDWI images for the three study areas. It was observed that the value distributions of the three types of images were quite different. The distribution of posterior probability values exhibits an obvious double-peak pattern, with most of the pixels having either low values (close to 0) or high values (close to 1) in all three study areas. Moreover, the double-peak pattern in the histograms of mNDWI and NDWI images was not significant. Since the thresholds are usually determined based on the histogram, it can be concluded that thresholds are easier to determine for the posterior probability maps than the other two as the difference between the two classes of pixels is much greater. As the distances between two peaks grow longer, and with a very limited number of pixels between the peaks, it can be inferred that segmenting the posterior probability image with a series of different thresholds would not generate many uncertainties. This suggests that the posterior probability method has higher stability than the traditional water index methods.

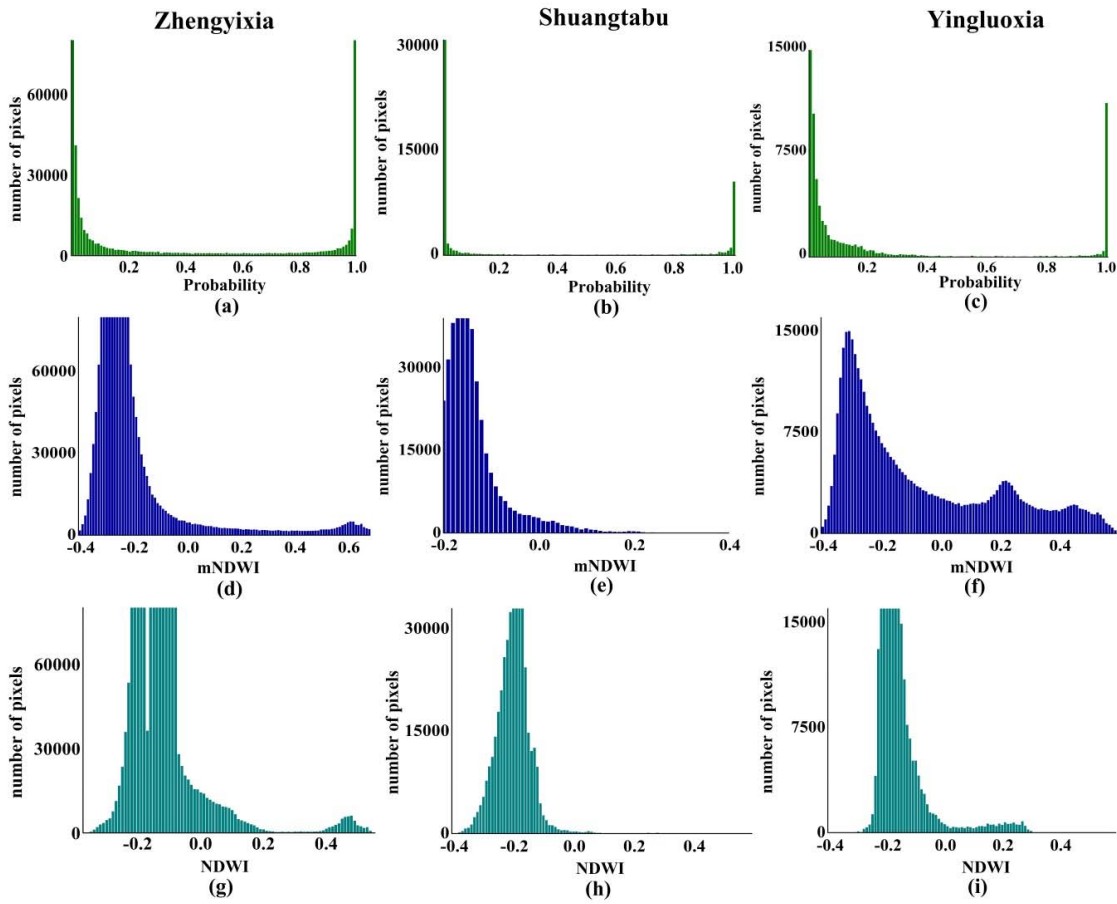

**Figure 8.** Histograms of posterior probability, modified Normalized Difference Water Index (mNDWI), and NDWI for the Zhengyixia, Shuangtabu, and Yingluoxia reaches.

An experiment was conducted by trialing a series of threshold values to segment the three types of images into binary water maps. The Sentinel-2 images were also employed for reference. The Kappa coefficient was calculated and used as the accuracy indicator. Figure 9 shows the Kappa variation as the threshold value changes on the posterior probability images, mNDWI, and NDWI images for all three study areas. It was observed that, while the threshold value changes from 0.30–0.98 on the posterior probability maps, the corresponding binary water maps for all the three reaches maintain a Kappa coefficient greater than 0.60. The accuracy increases steadily as the threshold value increases and the Kappa value reaches a high peak when the threshold is around 0.85 in the Zhengyixia reach, 0.96 in the Shuangtabu reach, and 0.56 in the Yingluoxia reach, while for the mNDWI and NDWI images, first-increase-then-decrease patterns were expected. In the beginning, when the threshold value increased, the Kappa coefficient also increased and reached the highest peak when the threshold was optimal. After that, the accuracy decreased as the threshold continued to increase. The variation of the Kappa coefficient can be significant as the threshold value changes. This means that the threshold must be optimized carefully, otherwise, the accuracy could be affected.

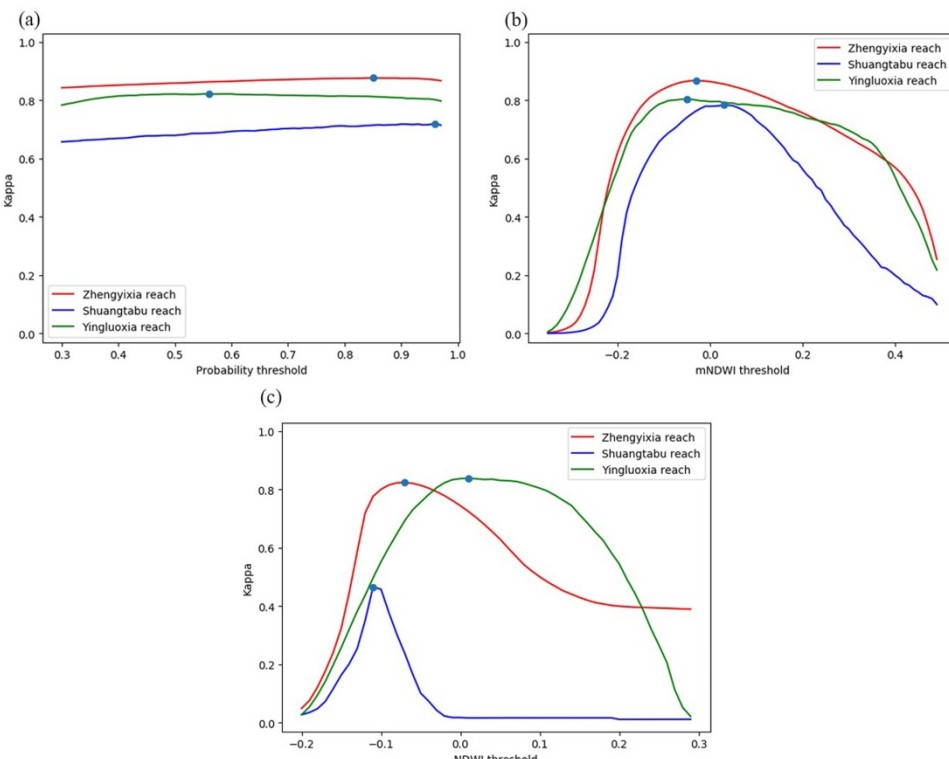

**Figure 9.** The Kappa coefficient values when using different thresholding values for (**a**) posterior probability images, (**b**) mNDWI images, and (**c**) NDWI images at three river reaches. Optimal threshold values were highlighted with dots on each line.

The threshold optimization process shown in Figure 9 is similar to the OTSU algorithm [27], except that the vertical axis shows inter-class variation instead of the Kappa coefficient in OTSU. This is a result of the ground truth data, such as the referencing Sentinel-2 map, not always being available. Under that circumstance, when optimizing the threshold for an unknown area, we must assume that the threshold that generates the smallest inter-class variation is the optimal threshold. Since we have a referencing water classification map, the threshold optimization process can be more effective and trustworthy, bearing in mind that we cannot usually find an optimal threshold without the reference.

*4.3. Binary Water Maps*

Using the optimized threshold values identified in Figure 10, binary water maps were derived from posterior probability images, mNDWI images, and NDWI images. Figure 10 shows the three binary maps in the Zhengyixia reach, along with the referencing Sentinel-2 water map. The demonstration zones of the posterior probability and mNDWI derived water maps appear to be consistent with the referencing water map, and some small water bodies were successfully restored. The NDWI derived map seems to miss some water pixels, especially those on the small tributaries. In order to quantify the accuracy of the three water maps, a series of accuracy indices, including overall accuracy, commission error, omission error, Kappa coefficient and CSI were calculated and listed in Table 3. Both PPSVM and mNDWI derived results demonstrated higher accuracy than NDWI. Both the commission and omission errors of PPSVM derived results were less than the results of the traditional water index methods.

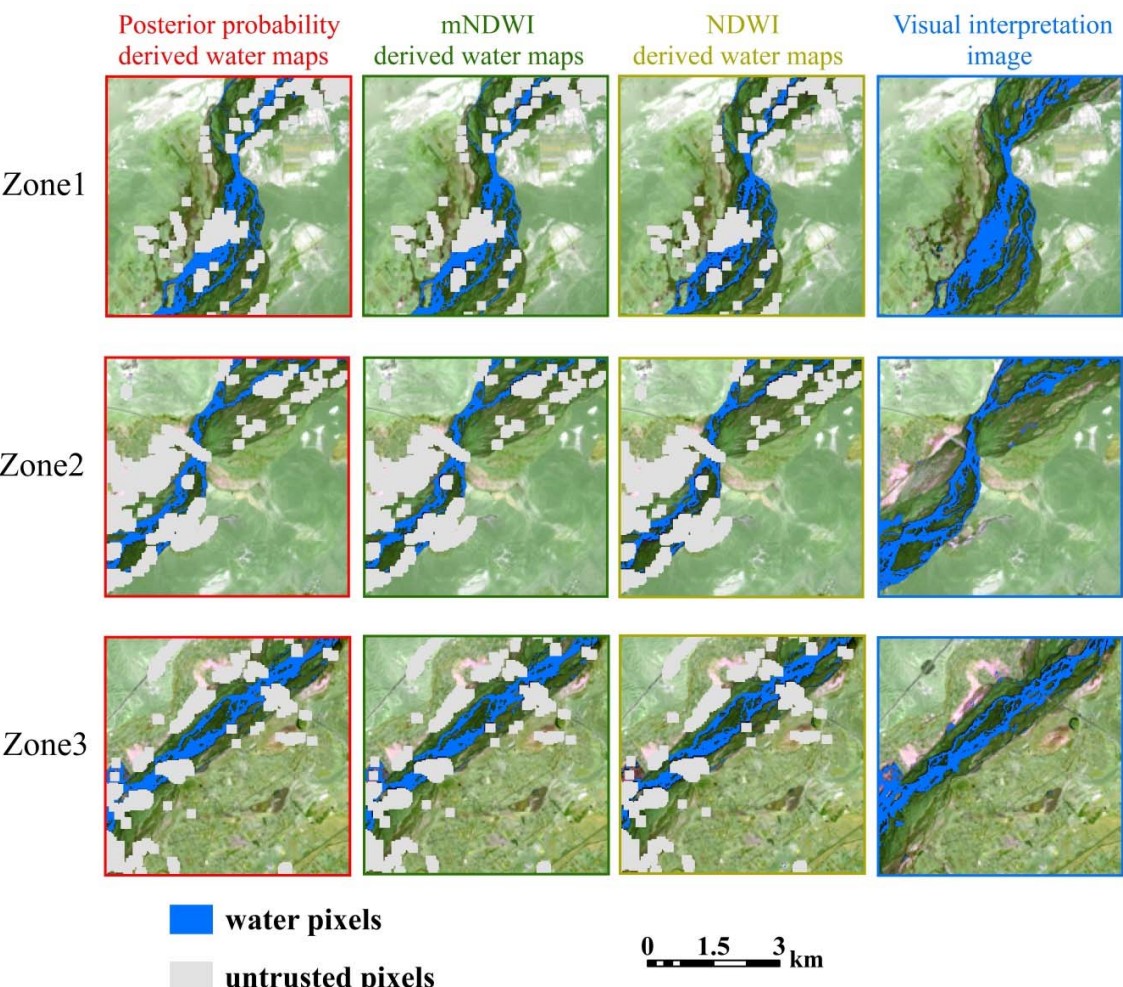

**Figure 10.** Binary water maps derived from posterior probability, mNDWI, NDWI, and Sentinel-2 visual interpretation of three demonstration zones in the Zhengyixia reach.

**Table 3.** Accuracy indices of three water maps derived from different methods for The Zhengyixia Reach.

| Method | Overall Accuracy | Commission Error | Omission Error | Kappa | Critical Success Index (CSI) |
|--------|------------------|------------------|----------------|-------|------------------------------|
| PPSVM | 98.3% | 0.9% | 0.6% | 0.877 | 0.795 |
| mNDWI | 98.2% | 1.0% | 0.7% | 0.868 | 0.781 |
| NDWI | 97.7% | 1.1% | 1.2% | 0.824 | 0.723 |

Figure 11 and Table 4 are the resulting maps and accuracy indices for the Shuangtabu reach. The binary water map derived from the proposed PPSVM appears to be similar to the referencing map, while the NDWI-derived map has more commission and omission errors. This is also clear from Table 4, where the mNDWI method derives the highest Kappa, CSI and overall accuracy. The proposed PPSVM method had similar accuracy as mNDWI, while NDWI showed the worst accuracy among the three methods.

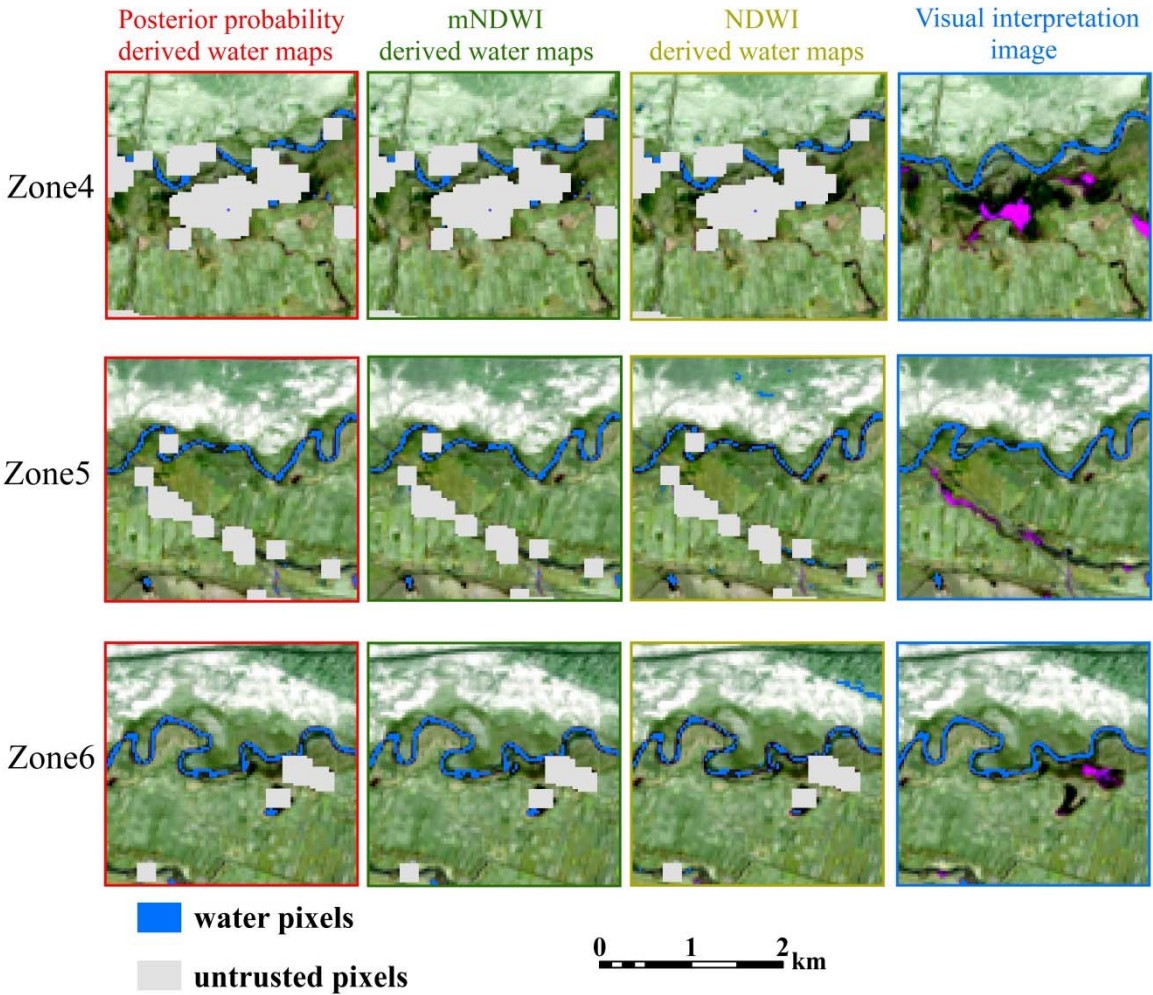

**Figure 11.** Binary water maps derived from posterior probability, mNDWI, NDWI, and Sentinel-2 visual interpretation for the three demonstration zones in the Shuangtabu reach.

**Table 4.** Accuracy indices of three water maps derived from different methods for the Shuangtabu Reach.

| Method | Overall Accuracy | Commission Error | Omission Error | Kappa | Critical Success Index (CSI) |
|---|---|---|---|---|---|
| PPSVM | 98.9% | 0.6% | 0.5% | 0.719 | 0.574 |
| mNDWI | 99.1% | 0.5% | 0.3% | 0.784 | 0.650 |
| NDWI | 97.6% | 1.5% | 0.9% | 0.465 | 0.321 |

Figure 12 and Table 5 are the resulting maps and accuracy indices for the Yingluoxia reach. This is a reach that belongs to the middle and upstream of the Heihe River, located in a mountainous area. Hill shade is a persistent issue that frequently intervenes in water mapping. However, since we have optimized our thresholds with a ground truth water map, it can be observed from Figure 12 that the results of all three methods were reasonable and unaffected. The overall accuracy, CSI and Kappa of these methods were similar, with overall accuracy over 98%, CSI around 0.7 and Kappa values greater than 0.8.

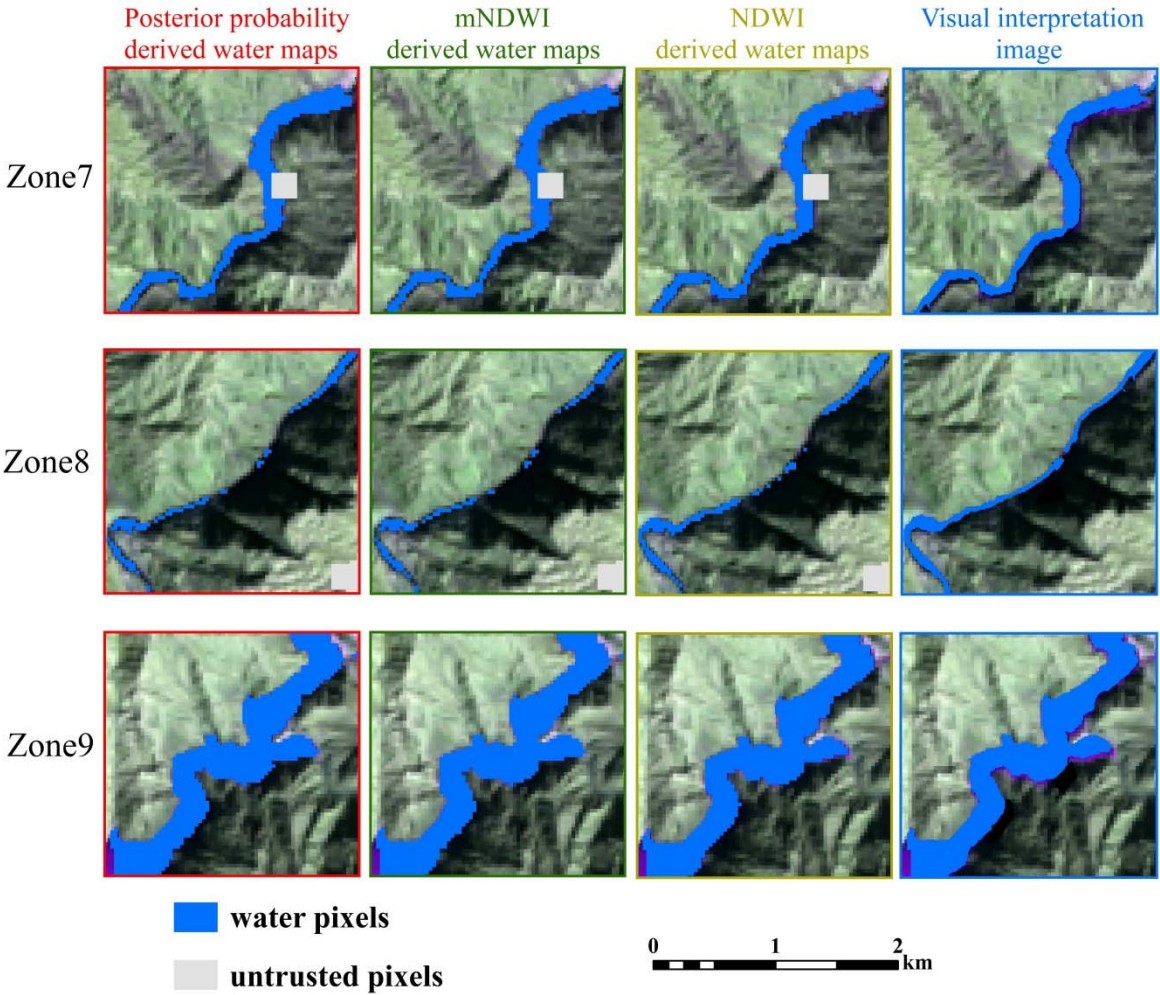

**Figure 12.** Binary water maps derived from posterior probability, mNDWI, NDWI, and Sentinel-2 visual interpreted for three demonstration zones in the Yingluoxia reach.

**Table 5.** Accuracy indices of three water maps derived from different methods for the Yingluoxia Reach.

| Method | Overall Accuracy | Commission Error | Omission Error | Kappa | Critical Success Index (CSI) |
|---|---|---|---|---|---|
| PPSVM | 98.8% | 0.6% | 0.6% | 0.822 | 0.707 |
| mNDWI | 98.6% | 0.7% | 0.7% | 0.804 | 0.682 |
| NDWI | 98.9% | 0.5% | 0.6% | 0.839 | 0.730 |

In order to further showcase the potential of the three methods in delineating water pixels and hill shade pixels, we manually identified all the hill shade pixels from the color composite image, then made boxplots to demonstrate the value distributions of posterior probability, mNDWI, and NDWI (Figure 13). The blue dots in Figure 13 represent the optimal thresholds identified in Figure 9. In terms of the posterior probability image, hill shade pixels have probability values ranging from 0–0.38, while the optimal threshold is 0.56, which is far away from the value range of the hill shade pixel. This suggests that the hill shade pixels are almost unlikely to be identified as water pixels based on the PPSVM method. On the contrary, optimal thresholds for the mNDWI and NDWI images are not as far away from value ranges of the hill shade pixels, meaning that some hill shade pixels may be incorrectly identified as water pixels if the threshold changes. This confirms that the proposed PPSVM method has a better performance for river water mapping in mountainous areas.

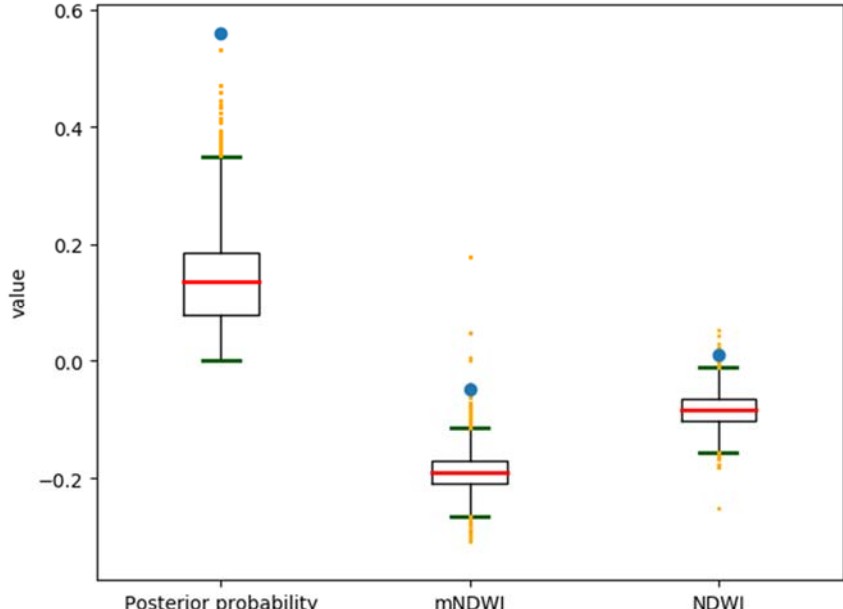

**Figure 13.** Boxplots of posterior probability, mNDWI, and NDWI values of hill shade pixels in the Yingluoxia reach (blue dots represent the optimal thresholds).

## 5. Conclusions

This study proposes a posterior probability SVM method that takes five water-sensitive Landsat OLI bands and a terrain index as the input to map river water bodies while maintaining the water-mapping uncertainties using probability values. It has to be noted that although some high-resolution images, such as GF-1/2, Worldview, etc., usually have no corresponding bands as were used here, we can also apply similar spectral analysis to find their appropriate bands, or simply select all available bands as the input to construct the feature set for the SVM classifier. It is just that the classification results might possibly not be as good when the SWIR band is missing.

Initially, FMask was applied to identify cloud coverage and cloud shadow pixels, which broadly exist on optical images. These pixels were marked as untrusted pixels and were not classified as either water or non-water. Although the untrusted pixels have not been discussed in this work, they could be useful for blending multisource river water mapping results to derive a more accurate and trustworthy water map later. Furthermore, FMask has another contribution, which is generating reliable pure water and pure land samples for training our PPSVM model.

Compared with traditional water index methods that enhance water and land contrast by combing several bands into a water-sensitive index, the proposed PPSVM identifies water pixels based on a support vector between water and land samples in terms of several carefully selected features. By adding a terrain index, HAND, into the feature set, our PPSVM method was able to utilize terrain characteristics to avoid confusion caused by hill shade. It is also noted that the adoption of a HAND image derived from SRTM data would also introduce some uncertainties. This may be due to the coarse resolution and moderate accuracy of SRTM DEM data. Besides, HAND index can be problematic for very flat areas.

Using visually interpreted Sentinel-2 image for reference, the proposed method proved able to produce reliable and accurate water-mapping results, even though we did not find significant accuracy improvement compared with the traditional mNDWI and NDWI methods. This is largely due to the thresholds being optimized by referencing a water map, which leads to all of the resulting maps having high accuracy. If there is no referencing map available, as it usually is, the accuracy difference between the three methods would be amplified. Our proposed PPSVM method would exhibit its advantage in the stability of thresholding, as was demonstrated in Section 4.2. Therefore, we would

like to summarize the advantage of our proposed algorithm to be more stable, and able to quantify uncertainty, compared with traditional index methods.

The proposed method can be considered as a unified framework that takes remote-sensing images and terrain index image as input for extracting river water area and quantifying the uncertainties. Therefore, even though this study uses Landsat OLI as the major input, the proposed method is also applicable to Sentinel-2 images with almost no alteration. Further study would be to test this model for Sentinel-1 SAR images, which would then facilitate a fusion study that uses Sentinel-1, Sentinel-2, and Landsat-8 together for enhanced river water mapping. Untrusted cloud-affected pixels can be confirmed by SAR image, and integrating these three data sources would enable a high temporal resolution for up to 3–5 days. This would further benefit related studies, including highly insensitive river flood monitoring and river discharge estimation.

**Author Contributions:** Q.L.: Methodology, Software, Writing—Original draft; C.H.: Conceptualization, Methodology, Writing—Reviewing and Editing, Supervision; Z.S.: Data curation, Validation, Visualization; S.Z.: Resources, Funding acquisition. All authors have read and agreed to the published version of the manuscript.

**Funding:** This research was funded by the National Key R&D Program of China, grant number 2017YFC0404302, and the National Natural Science Foundation of China, grant number 41501460.

**Acknowledgments:** The authors would like to thank NASA, ESA, and USGS for providing Landsat-8, Sentinel-2, and SRTM DEM data for this study. The authors want to thank Enago (www.enago.cn) for providing English polish to our manuscript. We would like to thank three anonymous reviewers for their constructive comments that help us significantly improve our paper.

**Conflicts of Interest:** The authors declare no conflict of interest.

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
