# Peer review of "Probabilistic River Water Mapping from Landsat-8 Using the Support Vector Machine Method"

_remotesensing, doi:10.3390/rs12091374_

Round 1

Reviewer 1 Report

To the authors, this paper for the most part was well-written and timely in nature.  Overall, I thought that this was a very good attempt by the authors to provide a Support Vector Machine (SVM)-based river water mapping method that can quantify the extraction uncertainties simultaneously utilizing Landsat-8 imagery.   Thus, providing valuable insights of how it was found that the proposed SVM method achieved generally high accuracy with a weighted root mean square difference of less than 0.1.   Finally, the proposed method resulted in this method outperforming the traditional water index methods in terms of river mapping accuracy and thresholding stability. With that in mind, I would have to request that this manuscript undergo a few revisions before publication within the Remote Sensing journal.  Here are my comments that I have regarding some of the information that was presented within this submitted manuscript.

Overall Comment: The FMask Algorithm, NDWI, mNDWI, overall accuracy, commission error, omission error, and the Kappa coefficient are all either algorithms, indices, or accuracy assessments that are utilized and discussed within the body of the manuscript. Thus, they are a major portion of the information described within the methodology, results, and discussion sections of this manuscript.  However, none of their equations or formulas are found within the manuscript and should be included and explained for readers of this manuscript that are not familiar with remote sensing and terminology within imagery analysis.  With that in mind, at various sections within the manuscript where these algorithms, indices, and accuracy assessments are introduced, the authors need to add in their equations and/or formulas based upon my reasoning given above.  Once this is completed, I have no other comments regarding this manuscript.

Author Response

To the authors, this paper for the most part was well-written and timely in nature. Overall, I thought that this was a very good attempt by the authors to provide a Support Vector Machine (SVM)-based river water mapping method that can quantify the extraction uncertainties simultaneously utilizing Landsat-8 imagery. Thus, providing valuable insights of how it was found that the proposed SVM method achieved generally high accuracy with a weighted root mean square difference of less than 0.1. Finally, the proposed method resulted in this method outperforming the traditional water index methods in terms of river mapping accuracy and thresholding stability. With that in mind, I would have to request that this manuscript undergo a few revisions before publication within the Remote Sensing journal.  Here are my comments that I have regarding some of the information that was presented within this submitted manuscript.

Response: Thank you very much for your recognition and helpful comments. We have revised our manuscript carefully according to your comments. Below are the responses.

The FMask Algorithm, NDWI, mNDWI, overall accuracy, commission error, omission error, and the Kappa coefficient are all either algorithms, indices, or accuracy assessments that are utilized and discussed within the body of the manuscript. Thus, they are a major portion of the information described within the methodology, results, and discussion sections of this manuscript. However, none of their equations or formulas are found within the manuscript and should be included and explained for readers of this manuscript that are not familiar with remote sensing and terminology within imagery analysis. With that in mind, at various sections within the manuscript where these algorithms, indices, and accuracy assessments are introduced, the authors need to add in their equations and/or formulas based upon my reasoning given above. Once this is completed, I have no other comments regarding this manuscript.

Response: We have added the algorithms used and discussed in the body of this article, including the formulas and equations of FMask algorithm, NDWI, mNDWI, overall accuracy, commission error, omission error, and the Kappa coefficient, and their corresponding explanations, for the sake of readers with broad backgrounds.

Reviewer 2 Report

This paper describes a supervised SVM application to classifying river waters in Landsat-8 imagery with implicit uncertainty representation. The paper is interesting in that it accounts directly for uncertainty in the mapping of water. The methodology is scientifically sound and the paper is generally well written. I have only a few minor to moderate comments:

  • The training model includes the SWIR bands which I think is a good choice but this means that the model would not be directly applicable to other optical sensors, namely S-2, which do not carry this band. The authors should comment on this;
  • The SRTM & HANDS are known to be problematic and would restrict this method to extract waters only linked to rivers and also would add considerable uncertainty themselves to the mapping - this should be commented on in the paper;
  • The overall accuracy measure is known to be biased towards overestimation and also much influenced by an exaggeration of the dry/dry score. I recommend the authors use also CSI to show performances in Table 3, along with the overall accuracy;
  • The resolution of the figures should be improved.

Author Response

This paper describes a supervised SVM application to classifying river waters in Landsat-8 imagery with implicit uncertainty representation. The paper is interesting in that it accounts directly for uncertainty in the mapping of water. The methodology is scientifically sound and the paper is generally well written. I have only a few minor to moderate comments:

Response: Thank you very much for your recognition and helpful comments. We have revised our manuscript carefully in accordance with your valuable comments. Below are our responses.

The training model includes the SWIR bands which I think is a good choice but this means that the model would not be directly applicable to other optical sensors, namely S-2, which do not carry this band. The authors should comment on this;

Response: We take SWIR bands as part of the inputs, based on our spectral analysis of water samples. SWIR bands have also been broadly proven to be effective in numerous studies. Indeed, there are some optical sensors, especially those high resolution sensors, such as GF-1/2, WorldView, have no SWIR bands, which makes our selected feature set not directly applicable. However, independent spectral analysis could still be carried out for these data to find their optimal bands and construct their own feature set for the PPSVM method. Most of the multispectral data, including Sentinel-2 actually, have SWIR channels (Band 11 & 12). It is just that the spatial resolutions of different types of bands might be a little bit different, which requires special treatment before these bands to be input into the PPSVM method. Actually, one of our further studies is to investigate how to fuse S-2 with Landsat for river water mapping for higher frequency and less uncertainty. We have added corresponding comments to the conclusion section to clarify this.

“It has to be noted that although some high resolution images, such as GF-1/2, Worldview, etc., usually have no corresponding bands as were used here, we can also apply similar spectral analysis to find their appropriate bands, or simply select all available bands as the input to construct the feature set for SVM classifier. It is just that the classification results might possibly be not as good when the SWIR band is missing.”

The SRTM & HANDS are known to be problematic and would restrict this method to extract waters only linked to rivers and also would add considerable uncertainty themselves to the mapping - this should be commented on in the paper;

Response: Indeed, HAND index derived from SRTM causes some problems due to its assumptions and data quality. It would be problematic for detecting non-river-induced inundations, especially in flat areas. In this study, since our major purpose is to detect river water bodies, we assume that terrain information was helpful for identifying river water, and take HAND image derived from SRTM DEM as a input feature for PPSVM method. It works fine so far in our case studies, except some subtle areas that were restricted by the coarse resolution of SRTM data. But sure, we agree it is worth mentioning the limitations about using HAND and SRTM for water detection. The corresponding comments have been supplemented to the Conclusion section.

“It is also noted that the adoption of HAND image derived from SRTM data would also introduce some uncertainties. This may due to the coarse resolution and moderate accuracy of SRTM DEM data. Besides, HAND index can be problematic for very flat areas.”

The overall accuracy measure is known to be biased towards overestimation and also much influenced by an exaggeration of the dry/dry score. I recommend the authors use also CSI to show performances in Table 3, along with the overall accuracy;

Response: Thank you for your suggestion, we have added the Critical Success Index (CSI) as a supplementary accuracy index for accuracy assessment, as shown in Table 3. According to the CSI values, the accuracy of our results is still good.

The resolution of the figures should be improved.

Response: We have improved the resolution of all our figures.

Reviewer 3 Report

Comments to the Author

Reviewer recommendation and comments for Manuscript ID: remotesensing-758285, entitled “Probabilistic River Water Mapping from Landsat-8 Using Support Vector Machine Method" for Remote Sensing.

General Comments:

In this study authors have proposed a Support Vector Machine (SVM) based algorithm named as PPSVM, for the estimation of river water extent. The proposed algorithm was tested on Landsat-8 Operational Land Imager (OLI) for three different dates. The results of PPSVM derived water extent were validated with the reference Sentinel-2 A/B images at 10 m spatial resolution. Also, the results of PPSVM were compared with two well-known water extent mapping algorithms i.e. Normalized Difference Water Index (NDWI) and modified Normalized Difference Water Index (mNDWI).

After going through the manuscript I have found following shortcomings in the manuscript

  • Authors have used Sentinel-2 A/B water maps as reference for the visual interpretation of their results. But, it is not clear in the manuscript that how these reference maps were generated? Were these reference maps obtained as a ready-to-use product or they were generated in-house?
  • Although authors have described a detailed methodology for their proposed algorithm but there is no significant differences in the water extent mapped by PPSVM and with NDWI or mNDWI. Why someone should use a very complex method (which authors have proposed) over the simple NDWI or mNDWI? I cannot see significant differences or the advantages of using PPSVM.
  • In figure 3, I am unable to understand the logic of drawing the surface reflectance and brightness temperature bands in a single figure. Do authors think that "surface reflectance" and "brightness temperature" are the same quantities and can be compared? Surface reflectance range from 0 to 1 or from 0% to 100% and it is unit less, while temperature has different ranges on different features and certainly have a unit either degree centigrade or Fahrenheit. Authors need to modify this figure.
  • A major English language editing is required.

My further comments are as follows;

  1. Page 1 line 13, "from remote sensing imagery often result in uncertainties" From this sentence it seems that authors are going to propose some other method for river water extent mapping, other than Remote Sensing. Authors need to rephrase this sentence.
  2. Page 1 line 23, "with traditional water indices methods" provide names of those water indices.
  3. Page 1 line 25, only one statistical metric (i.e. weighted root mean square difference) was used? I think it’s not enough to claim the good performance of the proposed SVM model. Therefore, authors need to use more statistical metrics to show the robustness of their model.
  4. Page 2 lines 72 to 78 belong to the Methodology section. Please move them.
  5. Page 2 line 80 "study area and data" but actually section 2 is about Materials and methods!
  6. Page 3 line 104, Landsat surface reflectance is distributed as Level-2 product not as Level-1. Please correct it. Reference: https://www.usgs.gov/land-resources/nli/landsat/landsat-surface-reflectance?qt-science_support_page_related_con=0#qt-science_support_page_related_con
  7. Page 3 line 105 "used as the input” input to what?
  8. Page 3 line 107, what was the level of Sentinel-2 A/B images i.e. was it used as TOA reflectance or surface reflectance? If surface reflectance, then who it was estimated?
  9. Page 4 lines 116 to 120, why FMask was used for identifying the clouds and cloud shadow pixels? Why authors not used the Quality Assurance (QA) band which is provided with Landsat-8 OLI imagery? This QA band also provide the pixel level information for the water and land.
  10. Page 4 Figure 2, STRM DEM should be written as SRTM DEM.
  11. Page 6 Figure 3, I am unable to understand the logic of drawing the surface reflectance and brightness temperature bands in a single figure. Do authors think that "surface reflectance" and "brightness temperature" are the same quantities and can be compared? Surface reflectance range from 0 to 1 or from 0% to 100% and it is unit less, while temperature has different ranges on different features and certainly have a unit either degree centigrade or Fahrenheit. Authors need to modify this figure.
  12. Section 3.5.2, I am concerned about this accuracy assessment method. The visual interpretation can be highly biased and it its results can vary from one user to another.
  13. Page 7 line 239, Please clarify what does it mean what authors say "Water maps"? Had authors obtained these water maps as a ready-made product or they produced them by themselves? If authors have produced them by themselves what was the methodology?
  14. Page 8 lines 245 to 247, again my question is about the generation of the Sentinel-2 water maps, how they were generated? I presume that authors have generated the Sentinel-2 water maps by themselves. In such case I feel that it does not make any sense to compare the results with Sentinel-2 water maps as both (Landsat and Sentinel-2) maps are generated by the authors.
  15. Figures 4, 5 and 6; what is the minimum and maximum width of the river sections which authors have studied in 9 different zones?
  16. Figures 10, 11 and 12; I do not see a significant difference between the binary water maps generated by the proposed method and the maps generated through mNDWI and NDWI. What is the advantage of using a much more complex method (which authors have proposed) over the simple mNDWI and NDWI which are easy to apply?

Author Response

In this study authors have proposed a Support Vector Machine (SVM) based algorithm named as PPSVM, for the estimation of river water extent. The proposed algorithm was tested on Landsat-8 Operational Land Imager (OLI) for three different dates. The results of PPSVM derived water extent were validated with the reference Sentinel-2 A/B images at 10 m spatial resolution. Also, the results of PPSVM were compared with two well-known water extent mapping algorithms i.e. Normalized Difference Water Index (NDWI) and modified Normalized Difference Water Index (mNDWI).

After going through the manuscript I have found following shortcomings in the manuscript

Response: Thank you very much for your constructive comments and careful corrections. We have revised the manuscript carefully according to them.

Authors have used Sentinel-2 A/B water maps as reference for the visual interpretation of their results. But, it is not clear in the manuscript that how these reference maps were generated? Were these reference maps obtained as a ready-to-use product or they were generated in-house?

Response: Validating maps of high dynamic objects such as water bodies is always painful. In this study, in order to validate the river water mapping results derived from 30 m resolution Landsat, we used water maps carefully visually-interpreted from 10 m resolution Sentinel-2 A/B images, assisted with high resolution Google Earth images. These referencing maps are unfortunately not readily available, and were generated in-house by ourselves. This has been clarified in the manuscript.

Although authors have described a detailed methodology for their proposed algorithm but there is no significant differences in the water extent mapped by PPSVM and with NDWI or mNDWI. Why someone should use a very complex method (which authors have proposed) over the simple NDWI or mNDWI? I cannot see significant differences or the advantages of using PPSVM.

Response: NDWI and mNDWI have been widely accepted as simple and effective methods. With proper threshold, they can achieve good results for most of the water bodies. The biggest issue for them is how to determine an optimal threshold value, which affects the accuracy significantly. In this study, we have given these two methods a lot of privileges, including adjusting their thresholds according to the referencing water maps. This is why their accuracy looks as good as ours. Nevertheless, in this study, we did not intent to prove that our method is definitely superior to the index methods regarding to the accuracy. The objectives of this study are to prove that our method is able to quantify the uncertainties while extracting river water bodies, and also, thresholding on our resultant probability maps is much easier because they have a much stable performance while threshold value changes. This means that variation in threshold values would introduce much less uncertainties, compared to the index methods. This has been further clarified in the manuscript.

In figure 3, I am unable to understand the logic of drawing the surface reflectance and brightness temperature bands in a single figure. Do authors think that "surface reflectance" and "brightness temperature" are the same quantities and can be compared? Surface reflectance range from 0 to 1 or from 0% to 100% and it is unit less, while temperature has different ranges on different features and certainly have a unit either degree centigrade or Fahrenheit. Authors need to modify this figure.

Response: Thank you for pointing this out, which is indeed a problem that we previously ignored. The main purpose of Figure 3 is to find out optimal bands with large difference in surface reflectance between water and land samples in order to construct the feature set for SVM classifier. Thermal bands are apparently out of the scope (and they actually does not belong to OLI data). Therefore, we have revised this Figure and Table 2 to eliminate all the brightness temperature related contents.

A major English language editing is required.

Response: We have carefully checked our language again, and employed a professional editing company (Enago) to check the English for us.

My further comments are as follows;

1. Page 1 line 13, "from remote sensing imagery often result in uncertainties" From this sentence it seems that authors are going to propose some other method for river water extent mapping, other than Remote Sensing. Authors need to rephrase this sentence.

Response: We have revised this sentence to "Traditional methods for river water mapping often result in significant uncertainties"

2. Page 1 line 23, "with traditional water indices methods" provide names of those water indices.

Response: revised.

3. Page 1 line 25, only one statistical metric (i.e. weighted root mean square difference) was used? I think it’s not enough to claim the good performance of the proposed SVM model. Therefore, authors need to use more statistical metrics to show the robustness of their model.

Response: We did use a series of different metrics to show the robustness of our method, which was presented in the manuscript. Here in the abstract, this description has been revised as below to convey more information about this.

"It was found that resultant probability maps obtained by the proposed SVM method achieved generally high accuracy with a weighted root mean square difference of less than 0.1. Other accuracy indices including Kappa coefficient and Critical Success Index also suggest that the proposed method outperformed the traditional water index methods in terms of river mapping accuracy and thresholding stability."

4. Page 2 lines 72 to 78 belong to the Methodology section. Please move them.

Response: revised.

5. Page 2 line 80 "study area and data" but actually section 2 is about Materials and methods!

Response: We have changed the title of section 2 as "Study Area and Data".

6. Page 3 line 104, Landsat surface reflectance is distributed as Level-2 product not as Level-1. Please correct it. Reference: https://www.usgs.gov/land-resources/nli/landsat/landsat-surface-reflectance?qt-science_support_page_related_con=0#qt-science_support_page_related_con

Response: Thanks for your correction, we have corrected this error in the manuscript.

7. Page 3 line 105 "used as the input”input to what?

Response: We have modified this sentence to "used as the input to SVM classifier".

8. Page 3 line 107, what was the level of Sentinel-2 A/B images i.e. was it used as TOA reflectance or surface reflectance? If surface reflectance, then who it was estimated?

Response: Since the Sentinel data were used only for visual interpretation, the readily available TOA product was acquired from GEE and used in this study. This has now been clarified as below.

"Sentinel-2A/2B Top Of Atmosphere (TOA) reflectance product from GEE acquired closely to Landsat-8 images (Table 1) were employed as the reference data source for validating the river mapping results, considering their relatively higher spatial resolution (10 m for several bands, including the near-infrared band). A strict visual interpretation was conducted carefully on Sentinel-2 images to generate reliable 10 m resolution water maps as the reference."

9. Page 4 lines 116 to 120, why FMask was used for identifying the clouds and cloud shadow pixels? Why authors not used the Quality Assurance (QA) band which is provided with Landsat-8 OLI imagery? This QA band also provide the pixel level information for the water and land.

Response: We noticed that there are two different versions of QA band, some was generated with a QA tool (https://www.usgs.gov/land-resources/nli/landsat/landsat-quality-assessment-tools?qt-science_support_page_related_con=0#qt-science_support_page_related_con), and some with FMask (https://www.usgs.gov/land-resources/nli/landsat/cfmask-algorithm). We chose FMask here simply because we believe this is a relatively new algorithm and has been used very widely recently. In fact, the QA band of Landsat product on GEE that we were using was also derived from FMask algorithm. We believe that the other versions of QA band are also applicable here, because the main purpose of using FMask here is to identify those cloud and cloud shadow pixels as untrusted pixels so that they will not cause confusion to our water mapping results. Another incidental advantage of applying FMask here is that it also produces confidence water and land samples in the meantime, which benefits our further process. But we believe that these samples could also be easily derived from other data sources if the other QA band was used instead of FMask.

10. Page 4 Figure 2, STRM DEM should be written as SRTM DEM.

Response: revised.

11. Page 6 Figure 3, I am unable to understand the logic of drawing the surface reflectance and brightness temperature bands in a single figure. Do authors think that "surface reflectance" and "brightness temperature" are the same quantities and can be compared? Surface reflectance range from 0 to 1 or from 0% to 100% and it is unit less, while temperature has different ranges on different features and certainly have a unit either degree centigrade or Fahrenheit. Authors need to modify this figure.

Response: Thanks. This figure has been revised to eliminate brightness temperature related contents.

12. Section 3.5.2, I am concerned about this accuracy assessment method. The visual interpretation can be highly biased and it its results can vary from one user to another.

Response: Validating maps of high dynamic objects such as water bodies is always painful. In this study, in order to validate the river water mapping results derived from 30 m resolution Landsat, we used water maps carefully visually-interpreted from 10 m resolution Sentinel-2 A/B images, assisted with high resolution Google Earth images. The visual interpretation indeed will be biased when it was conducted by different people. Here we have tried our best to ensure that the referencing water maps were generated as accurately as possible. For some particularly confusing areas, we also invited several other people to interpret independently in order to produce relatively high confidence water maps.

13. Page 7 line 239, Please clarify what does it mean what authors say "Water maps"? Had authors obtained these water maps as a ready-made product or they produced them by themselves? If authors have produced them by themselves what was the methodology?

Response: "Water maps" here refers to the "binary water maps", which is the binary water maps obtained by thresholding the water probability maps. We have revised the sentence to clarify this.

"A series of popular accuracy indicators, including overall accuracy, commission error, omission error, the Kappa coefficient [26], and critical success index (CSI) were employed to quantify the accuracy of these binary water maps obtained by probability images through thresholding."

14. Page 8 lines 245 to 247, again my question is about the generation of the Sentinel-2 water maps, how they were generated? I presume that authors have generated the Sentinel-2 water maps by themselves. In such case I feel that it does not make any sense to compare the results with Sentinel-2 water maps as both (Landsat and Sentinel-2) maps are generated by the authors.

Response: As we have responded in the former comments, it is always difficult to validate remotely sensed water product as the surface water changes drastically over space and time. Here, Sentinel-2 data were employed for visually interpreting water distribution considering that they have relatively higher spatial resolution, and are available on the close date of Landsat data. We have tried our best to make credible and objective water maps from Sentinel-2 data, and make this process completely independent with our experiment on mapping river water from Landsat using PPSVM. Therefore, we believe that our accuracy assessment results are reliable and objective.

15. Figures 4, 5 and 6; what is the minimum and maximum width of the river sections which authors have studied in 9 different zones?

Response: We have added the river width information in "study area" section.

" The width of these river sections varies from about 30 m to 100 m."

16. Figures 10, 11 and 12; I do not see a significant difference between the binary water maps generated by the proposed method and the maps generated through mNDWI and NDWI. What is the advantage of using a much more complex method (which authors have proposed) over the simple mNDWI and NDWI which are easy to apply?

Response: As we have responded in a former comment, the biggest issue for the traditional index method is how to determine an optimal threshold value, which affects the accuracy significantly. In this study, we have given these two methods a lot of privileges, including adjusting their thresholds according to the referencing water maps. This is why their accuracy looks as good as ours. Besides, in this study, we did not intent to prove that our method is definitely superior to the index methods regarding to the accuracy. The objectives of this study is to prove that our method is able to quantify the uncertainties while extracting river water bodies, and also, thresholding on our resultant probability maps is much easier because they have a much stable performance with different thresholds. This means that variation in threshold values would introduce much less uncertainties, compared to the index methods.

Round 2

Reviewer 3 Report

Notes;

Reviewer’s Comments/suggestion on manuscript version 1: Black Writing

Authors Responses: Green Writing

Reviewer’s Comments/suggestion on manuscript version 2: Red Writing

_____________________________________________________________Thanks for providing the revised version of the manuscript. I still have following concerns

  • Authors have mentioned they have created the Reference water maps at 10 m spatial resolution from Sentinel-2 A/B (S2AB) through visual interpretation and from Google Earth Imagery. But, still it is not clear how it was done. Have authors applied their proposed algorithm PPSVM for water maps?
  • Authors have emphasized that they had adjusted the threshold values for NDWI and mNDWI, don’t authors think this is something to produce the results according to their own desire?

  • Authors have compared the same products i.e. water maps generated from Landsat and S2AB. But for the generation of Landsat water maps authors have used the Landsat surface reflectance product (after atmospheric correction) data while for S2AB authors have just used the TOA reflectance which includes all the atmospheric perturbations. This is a serious point to be noted. The results from two products cannot be compared.

My further comments are as follows;

Authors have used Sentinel-2 A/B water maps as reference for the visual interpretation of their results. But, it is not clear in the manuscript that how these reference maps were generated? Were these reference maps obtained as a ready-to-use product or they were generated in-house?

Response: Validating maps of high dynamic objects such as water bodies is always painful. In this study, in order to validate the river water mapping results derived from 30 m resolution Landsat, we used water maps carefully visually-interpreted from 10 m resolution Sentinel-2 A/B images, assisted with high resolution Google Earth images. These referencing maps are unfortunately not readily available, and were generated in-house by ourselves. This has been clarified in the manuscript.

Still there are two shortcomings here, (i) there is no indication/methodological steps which were involved for the generation of the Sentinel-2 A/B reference water maps at 10 m spatial resolution in the manuscript. Have authors used the same methodology for the generation of S2AB water maps e.g. by applying the proposed PPSVM algorithm? (ii) It seems to be illogical to validate an algorithm with the self-generated reference water maps. I do not agree with this approach.

Although authors have described a detailed methodology for their proposed algorithm but there is no significant differences in the water extent mapped by PPSVM and with NDWI or mNDWI. Why someone should use a very complex method (which authors have proposed) over the simple NDWI or mNDWI? I cannot see significant differences or the advantages of using PPSVM.

Response: NDWI and mNDWI have been widely accepted as simple and effective methods. With proper threshold, they can achieve good results for most of the water bodies. The biggest issue for them is how to determine an optimal threshold value, which affects the accuracy significantly. In this study, we have given these two methods a lot of privileges, including adjusting their thresholds according to the referencing water maps. This is why their accuracy looks as good as ours. Nevertheless, in this study, we did not intent to prove that our method is definitely superior to the index methods regarding to the accuracy. The objectives of this study are to prove that our method is able to quantify the uncertainties while extracting river water bodies, and also, thresholding on our resultant probability maps is much easier because they have a much stable performance while threshold value changes. This means that variation in threshold values would introduce much less uncertainties, compared to the index methods. This has been further clarified in the manuscript.

Why authors had to adjust the threshold values for NDWI and mNDWI, don’t authors think this is something to produce the results according to their own desire? Authors are requested to report the results of NDWI and mNDWI without adjusting their threshold values, so their accuracy can be assessed better.

Page 3 line 107, what was the level of Sentinel-2 A/B images i.e. was it used as TOA reflectance or surface reflectance? If surface reflectance, then who it was estimated?

Response: Since the Sentinel data were used only for visual interpretation, the readily available TOA product was acquired from GEE and used in this study. This has now been clarified as below.

"Sentinel-2A/2B Top Of Atmosphere (TOA) reflectance product from GEE acquired closely to Landsat-8 images (Table 1) were employed as the reference data source for validating the river mapping results, considering their relatively higher spatial resolution (10 m for several bands, including the near-infrared band). A strict visual interpretation was conducted carefully on Sentinel-2 images to generate reliable 10 m resolution water maps as the reference."

What was the methodology for the generation of reference water maps from S2AB? Here authors have compared two same products i.e. water maps generated from Landsat and S2AB. But for the generation of Landsat water maps authors have used the Landsat surface reflectance (after atmospheric correction) data while for S2AB authors have just used the TOA reflectance which includes all the atmospheric perturbations. This is a serious point to be noted. The results from two products cannot be compared.

Author Response

There are some figures and tables in the response. Please see the attached file.

Round 3

Reviewer 3 Report

Thanks for providing the detailed responses to my comments. I have few minor suggestions;

  1. Provide all the figures and Table(s) (which authors have provided in the responses file) as supplementary material along with the manuscript.
  2. Add a statement in the manuscript that authors had used TOA reflectance data for S2AB for the delineation of water.